# Electrophysiological Correlates of Different Proactive Controls during Response Competition and Inhibition Tasks

**DOI:** 10.3390/brainsci13030455

**Published:** 2023-03-07

**Authors:** Marika Berchicci, Valentina Bianco, Hadiseh Hamidi, Linda Fiorini, Francesco Di Russo

**Affiliations:** 1Department of Movement, Human and Health Sciences, University of Rome “Foro Italico”, 00135 Rome, Italy; 2Department of Psychological, Humanistic and Territorial Sciences, University “G. d’Annunzio”, 66100 Chieti Scalo, Italy; 3Department of Brain and Behavioral Sciences, University of Pavia, 27100 Pavia, Italy; 4Department of Exercise and Health, University of Paderborn, 33098 Paderborn, Germany; 5IMT School for Advanced Studies, 55100 Lucca, Italy; 6IRCCS Santa Lucia Foundation, 00179 Rome, Italy

**Keywords:** ERPs, S-R mapping, motor readiness, visual activity, proactive control

## Abstract

The present study aims to investigate the behavioral outcomes and the antecedent brain dynamics during the preparation of tasks in which the discrimination is either about the choice (choice response task; CRT) or the action (Go/No-go), and in a task not requiring discrimination (simple response task; SRT). Using event-related potentials (ERPs), the mean amplitude over prefrontal, central, and parietal-occipital sites was analyzed in 20 young healthy participants in a time frame before stimulus presentation to assess cognitive, motor, and visual readiness, respectively. Behaviorally, participants were faster and more accurate in the SRT than in the CRT and the Go/No-go. At the electrophysiological level, the proactive cognitive and motor ERP components were larger in the CRT and the Go/No-go than the SRT, but the largest amplitude emerged in the Go/No-go. Further, the amplitude over parieto-occipital leads was enhanced in the SRT. The strongest intensity of the frontal negative expectancy wave over prefrontal leads in the Go/No-go task could be attributed to the largest uncertainty about the target presentation and subsequent motor response selection and execution. The enhanced sensory readiness in the SRT can be related to either an increased visual readiness associated with task requirements or a reduced overlap with proactive processing on the scalp.

## 1. Introduction

Discriminating between two possible alternatives could be either about a choice and the selection of different motor responses or about an action and the inhibition of a response. Behavioral outcomes and brain mechanisms underpinnings these kinds of discriminative processes have been studied in laboratory settings using different paradigms. Typical choice response time (CRT) tasks provide participants with multiple stimuli and each stimulus requires a different type of response (S-R mapping); for instance, participants must press a button with their right index finger when a red stimulus is displayed or press another button with their left index finger when a black stimulus is displayed [1]. Instead, when the discrimination is whether or not to make an action, the Go/No-go task is typically used; in this kind of paradigm, two stimulus categories are presented, and each category requires to either respond or withhold a response, e.g., [2,3,4,5]. For example, participants must press a button with their right index finger when a red stimulus is displayed, while they must prevent the response when a black stimulus is displayed. Cognitive experimental psychology settings have often compared discriminative tasks with simple response time tasks (SRT), wherein a single response is always required for any presented stimulus [3].

Behaviorally, the response time (RT) is usually much faster in simple than discriminative response tasks, and, within the discriminative tasks, the RT in the Go/No-go is faster than the CRT task; further, the accuracy of the responses is much higher and consistent in simple than discriminative response tasks [2,6,7,8]. Some earliest works [6,9] used subtractive methods to explain these behavioral results by assuming that the time of stimulus discrimination and response selection could be respectively estimated by subtracting the response time obtained in the SRT from those obtained in the Go/No-go, and in the CRT from the Go/No-go. This approach hypothesizes that common mechanisms would be canceled out, while extra processes will be enhanced; thus, choice procedures should include the response selection process in addition to the decision of interest. However, this methodology was questioned by a few authors [2,10], who applied quantitative models, such as the diffusion model [11], to support the view that the decision process is similar in two-choice procedure and Go/No-go tasks in the case of lexical decision paradigms [2].

The subtraction method was further applied to event-related potential (ERP) data by analyzing the lag between the onset of the lateralized readiness potential (LRP) and the production of the required response [7,8] to address the late motor processing invariance between tasks (simple, choice and Go/No-go tasks). However, the results were contradictory, possibly because of the inconsistencies among experimental designs (e.g., response uncertainty, responding hand). Danek & Mordkoff [8] questioned the idea of the motor stage invariance proposed by Miller & Low [7], suggesting that the Go/No-go task requires an additional inhibitory control process [12,13] that prevents participants from making false alarms by maintaining an equally high level of motor preparedness, which is lacking in the CRT. The suggestion that Go/No-go tasks require inhibitory processing during the preparation phase is nowadays well documented, e.g., [14,15,16]. However, traditional literature has mainly used a dichotomic view of motor preparedness/inhibition to elucidate and to compare the brain mechanisms of response competition (such as the CRT) and inhibition (such as the Go/No-go task).

The present study aims to investigate the behavioral outcomes and the task-set neural-related activity during the preparation of discriminative and simple tasks, trying to overcome the traditional dichotomic view. Indeed, the generation of expectation prior to observing stimuli requiring a specific action (or withholding it) is characterized by antecedent brain dynamics representing sensory, cognitive, and motor readiness interacting with each other, e.g., [17], and therefore not limited to motor and/or inhibitory processing.

To this end, we considered the following three anticipatory components and related brain mechanisms: (i) the movement readiness, such as the Bereischaftpotential (BP), likely originating in motor-related regions [18]; (ii) the proactive cognitive control, such as the prefrontal negativity (pN), likely originating in the inferior frontal gyrus (iFg) [4,19]; (iii) the sensory readiness, such as visual negativity (vN), likely originating in the related secondary sensory areas (e.g., extra-striate visual areas) [20,21].

Indeed, within the family of slow negative ERP waves occurring before stimulus presentation, a limited number of components have been described. The stimulus preceding negativity (SPN) is considered as an index of expectancy [22] in reward anticipation tasks, while, more generally, the contingent negative variation (CNV) entails different processes oriented to the sensory processing of the cue, the anticipation of the target and the motor preparation; for a review, see [23]. The BP [24] or readiness potential [25] is a brain correlate of action preparation largely investigated in ERP studies requiring motor response and voluntary movements, whose source has been localized in the supplementary motor area (SMA) and in the cingulate motor area (CMA) [4,26,27]. The BP has been documented in literature, triggering the ERP to either the motor emission or the stimulus presentation, although showing different features [3]. Beyond the mere motor preparation, there are further pre-stimulus components associated with cognitive functions. Accordingly, a slow negative wave has been described over prefrontal areas in tasks requiring a wide range of cognitive processes, named pN [4,18,28,29,30]. Previous studies combining functional magnetic resonance image (fMRI) and ERP techniques localized the pN source in the pars opercularis of the inferior frontal gyrus (iFg) [5,31]. Numerous studies showed that the pN component is modulated by several cognitive (and not only) factors, such as sustained spatial attention [32], individual response consistency [30], time on task and temporal expectancy [31], but also ageing [19]. Thus, the pN component reflects proactive cognitive control of the action, a form of anticipation and regulation of the behavior engaged proactively before the event [5,15,33,34,35,36,37]. It has been proposed that the BP and the pN may act in synchrony as a sort of accelerator/braking system [4,21,37] based on predictive internal models and acting on frontal-striatal circuits. Predictive (Bayesian) models suppose that upcoming events are perceptually anticipated, and a form of sensory task-specific anticipation can be observed in visual-motor tasks [38,39]. A potential ERP component related to sensory task-specific anticipation is the vN, recently reported in visual-motor tasks [20,21,32] and localized in extra-striate visual areas contralateral to the attended hemifield, supporting the hypothesis of a retinotopically specific shift of baseline activity in the visual cortex [40,41,42]. Thus, the vN may reflect a top-down signal, the allocation of preparatory attentional resources biasing cognitive processing in favor of stimuli at the attended location. Several studies proved the functional dissociation and interactions among the BP, the pN and the vN during task preparation and expectation [20,21,31,32,37].

The existence of an interplay among sensory, cognitive, and motor processing occurring during the preparation of tasks tapping into response selection (the ability to select the response to specific stimuli) and response inhibition (the ability to override a prepotent tendency to respond to specific stimuli) has never been reported in the literature.

In the present study, the selected visual stimuli are the same among tasks, as well as the inter-stimulus interval (ISI) and the stimulus frequency (*p* = 0.25), whilst they differ for the S-R mapping and the target probability. Instead, the S-R mapping is fixed and predictable in the SRT (4:1 S-R mapping) and the Go/No-go task (2:1 S-R mapping), but alternates in the CRT (two different 2:1 S-R mapping). Further, the target probability is higher in both the SRT and the CRT tasks, since it always requires a response, compared to the Go/No-go, which requires either responding or withholding the response. This variable manipulation should allow estimation of the parameter(s) that can potentially modulate the dynamics of the preparatory and anticipatory ERP correlates.

According to these premises, we hypothesized a similar activation at the sensory level and differences in cognitive-motor proactive control. Namely, the proactive cognitive control and the motor readiness should be reduced in the SRT compared to the CRT and the Go/No-go. Further, and more importantly, we hypothesized that, in the Go/No-go task, anticipatory motor and cognitive brain activities might be enhanced compared to the CRT, because of the need to inhibit a prepotent motor response before the stimulus presentation.

## 2. Materials and Methods

### 2.1. Participants

A priori power analysis for a within-subject repeated measure analysis of variance (ANOVA) design (G*Power 3.1.9.2) was performed to determine the minimum sample size. We selected a within-subject approach because it minimizes error variance associated with individual differences, keeping high statistical power despite the small sample size [43]. Since no previous ERP studies are available reporting the effect size of the interaction between the tasks used in the present study, a priori values for electrophysiological variables were estimated [44,45]. Using the automatic direct method available in G*Power, we selected a medium effect size f of 0.30, α error probability = 0.05, power (1-β error probability) = 0.80, number of groups = 1, number of measurements = 3, ε = 1. A minimum of 20 participants was required to obtain an actual power of 0.81. Thus, a total of 20 young (mean age ± SD: 22.7 ± 3.1 years) healthy participants (6 females) were involved in the study. All the participants were students at the University of Rome “Foro Italico” and they were given an extra credit for their participation at the study. Inclusion criteria were the following: right-handed [46], corrected-to-normal or normal vision. Exclusion criteria were the following: presence of neuropsychiatric disease, psychological disorders, and neurocognitive drug therapies. All provided written informed consent after a full explanation of the procedure by the experimenter, following the Declaration of Helsinki guidelines. The study was approved by the ethical committee of IRCCS Fondazione Santa Lucia (Rome, Italy).

### 2.2. Experimental Tasks and Procedures

Behavioral and electroencephalographic data were collected in a dimly lit and sound-attenuated room. Participants were comfortably seated 114 cm apart from a computer monitor, and their right hand was placed palm down on a button panel fixed on a board on their armchair. Visual stimuli were presented using Presentation^TM^ software (v.23, Neurobehavioral Systems, Inc. Berkeley, CA, USA) and consisted of four squared configurations (4 × 4°) made by vertical bars, horizontal bars, and a combination of both, which were displayed with equal probability (*p* = 0.25) for 250 ms. The four visual stimuli are shown in Figure 1. The inter-stimulus interval (ISI) varied from 1.5 to 2.5 s to avoid ERP overlapping between subsequent trials and to reduce anticipatory responses. The distribution of the inter-stimulus interval had a rectangular uniform shape; the frequency of occurrence was 5% of the total stimuli presented for each bin (both 50 and 100 ms). Each block lasted approximately 2 min, each containing 80 stimuli. The order of presentation was randomized within blocks. Participants were granted a pause whenever they needed it. At the center of the screen, a fixation point (yellow circle, diameter 0.15° of visual angle) on a black background was always displayed (please see Figure 1 for a schematic representation of the task).

Participants performed the simple response time task (SRT), the choice response time task (CRT) and the Go/No-go task on different days within a week; the order of tasks was counterbalanced between subjects using a balanced Latin square design to reduce the carryover effect of the within-subject design [47]. Participants received a short task training to familiarize themselves with stimulus categories and tasks.

The SRT consisted of 4 blocks, allowing a total of 320 trials. Participants were instructed to pay attention to the fixation point, to respond as quickly as possible to any stimulus (4 stimuli; *p* = 0.25) by pressing the button with their right index finger and to avoid anticipation errors and omissions.

The CRT consisted of 8 blocks, allowing a total of 640 trials (320 index finger responses and 320 middle finger responses). Participants were instructed to pay attention to the fixation point, to be very accurate (avoiding anticipation, omission, and commission errors), to press the button as quickly as possible with their right index finger when the index-target was displayed (2 stimuli; *p =* 0.5) and to press the button as quickly as possible with their right middle finger when the middle-target was displayed (2 stimuli; *p =* 0.5).

The Go/No-go task consisted of 8 blocks, allowing a total of 640 trials (320 targets and 320 non-targets). Participants were instructed to pay attention to the fixation point, to be very accurate (avoiding anticipation and commission errors), and to press the button as quickly as possible with their right index finger when the target was displayed (2 stimuli; *p =* 0.5) and to withhold their response when the non-target appeared (2 stimuli; *p =* 0.5).

Please note that index-targets of the CRT and targets of the Go/No-go were the same, as well as middle-targets of the CRT and non-targets of the Go/No-go. This procedure allowed a direct comparison of both behavioral and electroencephalographic data underpinning index finger responses among all tasks.

### 2.3. Behavioral Data: Analysis and Statistics

Trials with response anticipations (i.e., responses within 150 ms from stimulus onset) were excluded from further analyses in all tasks. At the group level, the mean response time (RT) for correct responses was calculated for each task. At the individual level, the median was calculated, because this was more stable than the mean. Accuracy was measured by the percentage of omission errors (OM%; i.e., missing responses to target stimuli) and commission errors (CE%; i.e., responses to non-target stimuli in the Go/No-go and responses with the wrong finger in the CRT). Responses slower than 1000 ms were excluded from the analyses because they were considered outliers. Further, to test the RT variability, the individual coefficient of variation (ICV) was calculated as the standard deviation/mean of the individual RT.

After being assured that data did not violate the assumption of normality and homoscedasticity (Shapiro-Wilk’s W and Levene’s tests), for the CRT, a paired t-test between the index finger and middle finger responses was performed for each behavioral variable and, these tests being non-significant (t_19_ < 1.35; p_s_ > 0.190), only the index finger was considered. Behavioral variables were then separately submitted to repeated measures analysis of variance (RMs-ANOVA) with Task (SRT, CRT, Go/No-go) as factor. The partial eta squared (η^2^_P_) and the Cohen’s d (d) were used to measure the effect size of the significant effects in the ANOVAs and in the t-tests, respectively. Post-hoc analyses were performed using the Bonferroni correction. To assess the reliability of the behavioral data (RT, ICV, OM%, and CE%), the odd-even split-half procedure was used. For each measurement, the score was first calculated based on the odd trials and then based on the even trials, making sure that each score was based on an equal number of trials. The split-half reliability was calculated by correlating these two scores and by applying a Spearman-Brown correction. The significance of each correlation coefficient was tested with ANOVAs comparing the correlation slope with zero. The overall alpha value was fixed at 0.05.

### 2.4. Electroencephalographic (EEG) Recording, Analysis, and Statistics

EEG was continuously recorded with BrainVision Recorder 1.2 using three BrainAmp^TM^ amplifiers, two of them connected to 64-active sensors ActiCap; data were processed using Analyzer 2.2 software (all by BrainProducts GmbH., Munich, Germany). Electrodes were mounted according to the 10-10 International System and referenced to averaged M1-M2 electrodes (which were placed on the left and right mastoids, respectively). EEG data were amplified, digitized at 250 Hz, band-pass filtered using a Butterworth zero-phase filter (0.01–30 Hz; second order) and stored for off-line analyses. Eye movements were monitored by electro-oculogram (EOG) recorded by the third BrainAmp amplifier (ExG type) in bipolar modality. Horizontal EOG was recorded with electrode pair over the left and right outer canthi of the eyes, while vertical EOG (VEOG) were recorded with an electrode pair below and above the left eye. Electrode impedances were kept below 5 KΩ.

Blink and vertical eye movement artefacts were automatically corrected by means of the independent component analysis [48]. VEOG-free data were submitted to a semi-automatic artefact rejection, excluding EEG with amplitudes exceeding the threshold of ±70 µV and trials with still horizontal eye movements.

To evaluate the pre-stimulus activity, EEG was segmented into 2000 ms epochs, starting 1100 ms before and ending 900 ms after stimulus onset; grand-average ERPs were obtained for each task, and then a baseline of 200 ms (−1100/−900 ms, in which the signal was flat and stable, according to previous studies; please see [3]) was applied. As mentioned in the Experimental tasks and Procedures section, to allow a direct comparison of electroencephalographic data underpinning index finger responses among all tasks, the pre-stimulus analyses were based on the average of trials requiring the response (Go trials) for the Go/No-go and the index finger response for the CRT, whereas all the trials were averaged for the SRT, reaching a comparable signal-to-noise ratio.

The statistical analysis focused on the last 600 ms before the stimulus onset, in a timeframe where the BP, the pN and the vN were previously detected ([4] for normative data). To select the regions of interest (ROIs), the collapsed localizers method was used [49]. Following this procedure, the pN was calculated using Fp1, Fp2, Fpz, and AFz sites for the prefrontal ROI (e.g., mean activity in the selected electrodes); the BP was calculated using Cz, CPz, and Pz sites for the central ROI; the vN was calculated using PO7 and PO8 for the parietal-occipital ROI. For statistical purposes, the 600 ms time window preceding the stimulus onset was divided into two sub-windows each lasting 300 ms: −600/−300 ms, and −300/0 ms. The mean ERP activity in the selected time windows of each ROI was separately submitted to a RM-ANOVA with Task (SRT, CRT, Go/No-go) as factor.

After testing for normality and homoscedasticity, ROIs amplitude of the considered components were submitted to paired t-tests between index-finger and middle-finger trials for the CRT and, this test being non-significant (t_19_ < 1.22; p_s_ > 0.230), only the index finger was considered. For the Go/No-go, only target trials were considered. Afterward, RM-ANOVAs were separately performed for each component using Task (SRT, CRT, Go/No-go) as factor. The partial eta squared (η^2^_P_) and the Cohen’s d (d) were used to measure the effect size of the significant effects in the ANOVAs and in the t-tests, respectively. Post-hoc analyses were performed using the Bonferroni correction; for significant interactions, the mean difference (mean dif) and the standard error (SE) were reported to provide information on the relative effect size. The overall alpha value was fixed at 0.05.

To visualize the ERP topography, spherical spline maps were rendered using BrainVision Analyzer 2.2 tools and were visualized with a top-flat view 120° wide. To better identify the pre-stimulus components, also the current source density (CSD) maps were calculated and displayed because they offer the advantage of reducing the volume conduction at the scalp level.

## 3. Results

This RM-ANOVA on the RT showed a significant Task effect; post-hoc analysis revealed that participants were faster in the SRT than the Go/No-go (*p* < 0.001, mean dif = 280.53, SE = 21.16) and the CRT (*p* < 0.001, mean dif = −225, SE = 21.16), and slower in the CRT than the Go/No-go (*p* = 0.045, t = 2.50, mean dif = 55, SE = 21.16).

A significant Task effect was also observed for the ICV; post-hoc analysis showed that the behavioral performance of the participants was less stable in the CRT than the Go/No-go (*p* = 0.037, mean dif = 0.038, SE = 0.014) and the SRT (*p* < 0.001, mean dif = 0.068, SE = 0.014).

The accuracy was measured as a percentage of omission and commission errors. With respect to OM%, a significant effect was observed, with similar performance between the CRT and the Go/No-go tasks and a minimal percentage of omission in the SRT compared to the CRT (*p* = 0.018, mean dif = 0.82, SE = 0.28) and the Go/No-go task (*p* = 0.030, mean dif = −0,77, SE = 0.014). Given the task requirements, CE% was compared only between the CRT and the Go/No-go, without showing significant differences. Please see Table 1 and Table 2 for behavioral data and statistical results, respectively.

Results of the odd-even split-half reliability tests showed that the correlations were high and significant for all behavioral measures (*r* > 0.89, *p* < 0.001).

For the pre-stimulus ERP activity, three main components have been investigated: the pN over the prefrontal cortex and represented by the prefrontal pool of electrodes (PF-ROI), the BP over motor-related region and depicted in the figure with the central-medial pool of electrodes (C-ROI), and the vN over visual brain regions and displayed in the figure with the parietal-occipital pool of electrodes (PO-ROI).

Inspection of the waveforms in Figure 2 shows that, at approximately 800 ms before stimulus onset, the vN rises almost concomitantly in all the tasks over the PO-ROI. Although during the first 400 ms the amplitude appears larger in the SRT than in discriminative tasks, later the amplitude reaches almost the same values for the SRT and the Go/No-go, while it is smaller in the CRT. At approximately 700 ms before stimulus onset, the BP over the C-ROI shows a steep increase in amplitude reaching its maximum right after stimulus onset. The BP amplitude is larger in the Go/No-go than in the CRT and the SRT. Finally, the pN over the PF-ROI starts at approximately 600 ms before stimulus onset only in the CRT and the Go/No-go, with the latter showing the largest amplitude.

Figure 3 shows both voltage and CSD maps referring to the two temporal windows analyzed (−600/−300 ms on the left part of the figure and −300/0 ms on the right part of the figure) in the three tasks. The maps in the first temporal window mainly show a focused central-parietal negativity, reflecting the vN and the BP. The maps for the second temporal window also show the pN over prefrontal sites. The maps in both time windows show stronger activity in the Go/No-go than the CRT and the SRT, especially for the pN and the BP. The components are more clearly visible in the CSD maps. It is worth noting that the topographical differentiation between the BP and the vN and their source localization has been previously performed using a subtractive approach (see [20,21]).

RM-ANOVAs were separately performed for the two temporal windows. With respect to the −600/−300 ms temporal window, a significant Task effect was observed for the pN component; post-hoc analysis revealed that the pN was smaller in the SRT than the Go/No-go (*p* = 0.003, mean dif = −1.02, SE = 0.25) and the CRT (*p* = 0.016, mean dif = −0.87, SE = 0.25). A significant Task effect was also found for the BP component; post-hoc showed that the BP was larger in the Go/No-go than the SRT (*p* = 0.004, mean dif = −0.84, SE = 0.24). No significant effects emerged for the vN component.

With respect to the −300/0 ms temporal window, a significant Task effect was observed for the pN component; post-hoc analysis revealed that the pN amplitude was reduced in the SRT compared to the Go/No-go (*p* < 0.000, mean dif = −1.55, SE = 0.38) and the CRT (*p* < 0.000, mean dif = −2.02, SE = 0.38). A significant Task effect was observed for the BP component; post-hoc analysis showed that the BP amplitude was larger in the Go/No-go than the CRT (*p* = 0.046, mean dif = 0.88, SE = 0.34) and the SRT (*p* = 0.009, mean dif = −1.10, SE = 0.34). A significant Task effect was also observed for the vN component; post-hoc analysis revealed that the vN amplitude was smaller in the CRT than the SRT (*p* = 0.027, mean dif = 0.80, SE = 0.31) and the Go/No-go (*p* = 0.040, mean dif = 0.85, SE = 0.31).

Furthermore, if the amplitude of the peak electrode (i.e., AFz) of the pN component is chosen as factor for the ANOVA, a significant difference across tasks is present, with post-hoc analysis showing larger amplitude in the Go/No-go (−2.08 ± 0.93 μV) than the CRT (*p* = 0.010; −1.34 ± 0.63 μV) and the SRT (*p* < 0.001; 0.53 ± 0.62 μV), and larger amplitude in the CRT than the SRT (*p* < 0.001). Please, see Table 3 for the pre-stimulus ERP data and Table 4 for statistical results.

## 4. Discussion

The main aim of this study was to investigate the task-set neural-related activations and the behavioral outcomes during the preparation of competition (CRT), inhibition (Go/No-go) and simple (SRT) tasks. Accordingly, we ought to test at what extent anticipatory electrophysiological activities are involved in the preparation of alternative response options differing in the type of S-R mapping and target probability. Behaviorally, we confirmed previous studies [2,6,7,8], showing that participants were slower and less accurate in the CRT than the Go/No-go task, and faster and more accurate in the SRT than the CRT and the Go/No-go tasks.

It is well known that the motor outcome (i.e., RT) is related to motor preparation, with larger BP amplitude for faster RT [2,6,7,8]; for normative data see [21]. Accordingly, we have observed a larger BP amplitude in the Go/No-go than the CRT in the last 300 ms before the stimulus onset and the SRT in the entire time window, and a larger BP amplitude in the CRT than the SRT in the full preparatory stage. Consistently, not only the BP but also the pN was enhanced in the Go/No-go task.

Among the studied tasks, the hazard function was similar, and uncertainty about the occurrence of the target (i.e., press a button with the right index finger) was maximal in the Go/No-go task, whilst that related to S-R mapping was maximal in the CRT. The strongest intensity of the frontal negative expectancy wave (the pN) in the Go/No-go task could be attributed to the largest uncertainty about the target presentation and subsequent motor response selection and execution, as shown by the BP amplitude in this task compared to the other two tasks (since the Go/No-go task did not demand a motor response on every trial).

However, we should also consider that the Go/No-go task challenges inhibitory control mechanisms mainly within the proactive phase [5,15]. Indeed, the Go/No-go task is essentially a proactive inhibitory task, because participants are required to withhold their responses instead of stopping them, as in a stop-signal task. However, by moving this ratio of target and non-target (e.g., 80/20, 50/50, 20/80) and the inter-stimulus interval, the involved neural and cognitive processes would also significantly change. For example, modulating the proportion of Go/No-go stimuli, [29] has observed enhanced BP, faster RT, and higher CE in frequent than rare (12% vs. 88%) Go trials. Wessel [50] also showed that a slow-paced (inter-trial interval higher than 4 s) version of this task would modulate prepotent motor response and, consequently, inhibitory control. Here, we used a 50/50 ratio of target/non-target and a relatively short ISI (1.5–2.5 s), still keeping the task as inhibitory.

Present data suggest that proactive control includes both motor readiness and cognitive, perhaps inhibitory, control as functionally different and interactive processes, overcoming the dichotomic view of motor preparedness/inhibition proposed by traditional approaches. Further, the preparation seems to be mainly modulated by the uncertainty about the target presentation (Go trials).

It has been previously demonstrated that proactive control and the interplay between motor readiness and inhibition contribute to behavioral outcomes, such as the speed-accuracy trade-off [11,51], the performance consistency [30] and the ability to fix an error [52]. Some studies proposed the hold your horse model [35] or an accelerating-braking system [4,18], responsible for preventing the emission of a response until the stimulus is perceptually and cognitively elaborated and the appropriate action defined. Crucial support for this finding comes from the study by Burle et al. [53], who suggested the involvement of inhibitory processing in CRTs. The authors used an approach integrating three techniques (i.e., EEG, transcranial magnetic stimulation, and electromyography) to demonstrate that, in between-hand two-choice RT tasks, motor activation is required for the responding hand, while motor inhibition is required for the non-responding hand. This inhibitory process has been supposed to occur at the level of response programming based on the information provided by the imperative stimulus within the SMA. We have, instead, proposed that the inhibitory control is part of the cognitive control exerted by the iFg over the SMA. Furthermore, the inhibitory control is likely absent, and the motor readiness is reduced in the SRT compared to the CRT and the Go/No-go, in line with our hypothesis.

The uncertainty about the target presentation seems to have influenced the proactive cortical control more than the variability of the S-R mapping, which may play a role after stimulus onset and not during the proactive phase.

Interestingly, the present study supports the view that the proactive stage in visual sensory-motor tasks includes a sensory readiness that is modulated by visuospatial attention [21,32]. Indeed, directing attention to a target location and expecting the presentation of a visual stimulus in that location leads to a retinotopic increase of the baseline activity in striate and extra-striate visual cortices [41], known as the baseline shift index. This bias in the occipital cortex would be triggered by top-down frontal regions, the inferior parietal cortex, and the superior temporal cortex [42]. The existence of sensory anticipation has also been acknowledged in other sensory modalities, such as the auditory and the somatosensory modalities, with the pre-activation of sensory-specific brain regions [20]. The enhanced vN amplitude in the SRT 300 ms before stimulus onset could represent an increased visual readiness associated with the task requirements, since the simple task always requires a motor response regardless the stimulus presented. The vN amplitude could also be larger in the SRT than the CRT and the Go/No-go because there is reduced overlap with proactive processing at the scalp level.

## 5. Conclusions

To summarize, we showed that both proactive cognitive and motor control were enhanced in the CRT and the Go/No-go, with the Go/No-go task showing the largest ERP amplitudes, while sensory readiness was enhanced in the SRT. Overall, the amplitude of all the considered components in the CRT was reduced compared to the Go/No-go task.

In the present research, we have demonstrated the importance of studying the phase preceding both the presentation of the stimulus and the execution of the action, overcoming the classic dichotomous view of inhibition/action, and the typical approach to the analysis of the psychophysiological processes following the stimulus presentation [54].

A better understanding of the electrophysiological processes involved in the preparation before choosing among response alternatives and/or in the preparation to act or not might be relevant in clinical contexts related to movement disorders, as in the case of cerebellar damage [55] and clinical conditions where a failure in inhibition is usually reported [56], as well as in psychiatric conditions, such as attention-deficit hyperactivity disorders [57] and eating disorders [58], in order to improve the reliability of the assessment and to suggest coherent and tailored intervention strategies. Finally, we acknowledge the limited sample size as a potential limitation of the study.

Future ERP studies are needed to address to what extent the observed findings are replicated using the same tasks in different sensory modalities, but also throughout the application of trial-by-trial analysis and mathematical and statistical Bayesian models. Further, neuroimaging studies are called to disambiguate the contribution of preparatory motor and pre-motor activities to discrimination processes requiring the decision to choose among possible alternatives vs. prompting action and inhibition processes.

## Figures and Tables

**Figure 1 brainsci-13-00455-f001:**
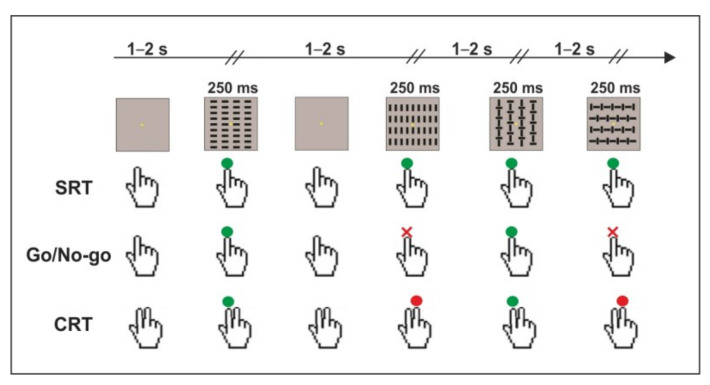
Schematic illustration of the three tasks: the simple response time task (SRT), the Go/No-go task, and the choice response time task (CRT). The green button corresponds to the response made with the index finger, the red button corresponds to the response made with the middle finger, and the red cross corresponds to the withheld response.

**Figure 2 brainsci-13-00455-f002:**
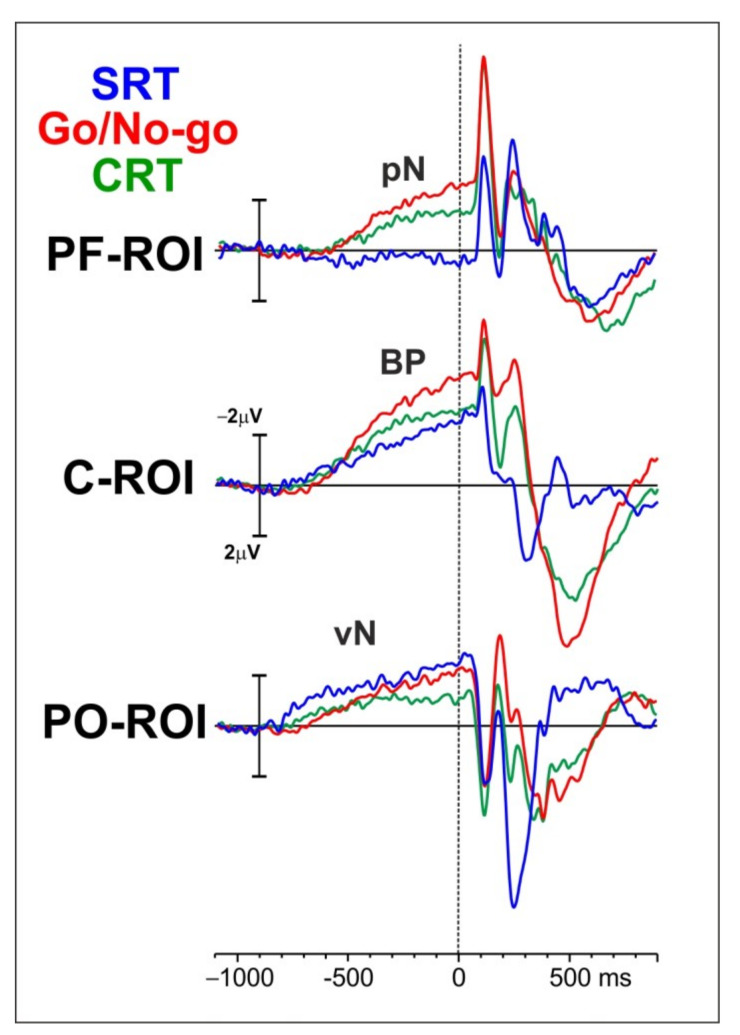
ERP waveforms for the three tasks (SRT in blue, Go/No-go in red, and CRT in green) and the three pools of electrodes (PF-ROI: prefrontal ROI; C-ROI: central ROI; PO-ROI: parietal-occipital ROI). The main components are depicted in the figure (BP: Bereitschaftspotential; pN: prefrontal negativity; vN: visual negativity).

**Figure 3 brainsci-13-00455-f003:**
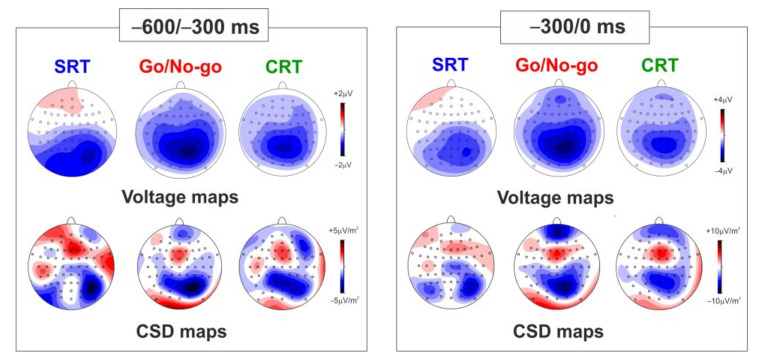
Voltage and CSD maps in the two time-windows: −600/−300 ms on the left side and −300/0 ms on the right side of the figure. The maps are shown for the three tasks (SRT, Go/No-go, and CRT).

**Table 1 brainsci-13-00455-t001:** Behavioral data (mean ± SD). RT: median response time in ms; ICV: intra-individual coefficient of variation = RT SD/RT Mean; OM%: percentage of missed responses; CE%: percentage of commission errors (response with the wrong finger in CRT or false alarms in Go/No-go).

	CRT		Go/No-Go		SRT
	Index Finger	Middle Finger	Go	No-Go	
RT	516 ± 109	527 ± 113	461 ± 91	408 ± 60	235 ± 53
ICV	0.22 ± 0.06	0.21 ± 0.06	0.18 ± 0.02	-----	0.15 ± 0.02
OM%	0.82 ± 1	1.27 ± 2.2	0.77 ± 1	-----	0.7 ± 0.5
CE%	9.90 ± 1.45	10.18 ± 1.72	9.26 ± 2.6	-----	-----

**Table 2 brainsci-13-00455-t002:** Statistical data for behavioral measures (Task effect). Degrees of freedom (DoF), Partial eta squared (η_p_^2^).

	DoF	F	*p*	η_p_^2^
RT	2, 38	94.5	<0.001	0.83
ICV	2, 38	11.8	0.001	0.38
OM%	2, 38	5.2	0.009	0.21
CE%	1, 19	0.5	0.822	

**Table 3 brainsci-13-00455-t003:** Pre-stimulus ERP activity (mean μV± SE) in the three tasks. The following components were considered: the pN (prefrontal ROI: Fp1, Fp2, Fpz, AFz), the BP (central ROI: Cz, CPz, Pz), and the vN (parietal-occipital ROI: PO7, PO8).

	CRT		Go/No-Go		SRT	
	−600/−300 ms	−300/0 ms	−600/−300 ms	−300/0 ms	−600/−300 ms	−300/0 ms
pN	−0.52 ± 1.1	−1.45 ± 1.5	−0.68 ± 1	−1.84 ± 1	0.34 ± 1	−0.08 ± 0.5
BP	−1.38 ± 0.6	−2.25 ± 1	−1.65 ± 0.6	−3.14 ± 1	−0.81 ± 1	−2.04 ± 1
vN	−0.73 ± 0.8	−0.95 ± 0.9	−0.84 ± 0.9	−1.55 ± 1	−1.22 ± 0.9	−1.80 ± 1

**Table 4 brainsci-13-00455-t004:** Statistical data of the ERP measures (Task effect). Degrees of freedom (DoF), Partial eta squared (η_p_^2^).

	DoF	F	*p*	η_p_^2^
pN (−600/−300)	2, 38	7.0	0.002	0.26
BP (−600/−300)	2, 38	6.2	0.004	0.24
vN (−600/−300)	2, 38	1.6	0.210	0.07
pN (−300/0)	2, 38	14.8	<0.001	0.43
BP (−300/0)	2, 38	5.5	0.007	0.22
vN (−300/0)	2, 38	4.8	0.014	0.20
pN peak	2, 38	66.5	<0.001	0.77

## Data Availability

Data are available on request.

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
