# Peer review of "Electrophysiological Correlates of Different Proactive Controls during Response Competition and Inhibition Tasks"

_brainsci, 2023, doi:10.3390/brainsci13030455_

Round 1

Reviewer 1 Report

Comments and Suggestions for Authors

In this manuscript, the authors compared three behavioral control task paradigms, namely simple response task, discrimination response task, and go/nogo task. They evaluated the electrophysiological correlations of different proactive controls. Overall, the manuscript is well written, and the procedures are carefully conducted. The reviewer just has several minor concerns which may help improve the manuscript.

1), abstract: the aim is too technical, can the authors explain the aim at the "research" level?

2), introduction: 1st sentence: the use of the term "choice" together with the example of choosing between cake and pizza let the reviewer expect (value-based) decision-making tasks. however, the study has nothing relevant to value or preference-based choices. this part should be revised.

3), figure 1: the abbreviations of SRT and CRT are better explained in the legend.

4), section 2.3 behavioral data analysis: why responses slower than 1000 ms are considered outliers?

5), tables 1 and 2: the statistical results reported in the main text are better also presented in the table, which is far easier to read.

Author Response

Comment: In this manuscript, the authors compared three behavioral control task paradigms, namely simple response task, discrimination response task, and go/nogo task. They evaluated the electrophysiological correlations of different proactive controls. Overall, the manuscript is well written, and the procedures are carefully conducted. The reviewer just has several minor concerns which may help improve the manuscript.

Authors’ reply: We thank the Reviewer for the positive consideration of the manuscript. We have addressed all the points raised by the Reviewer, hoping that the revised version of the manuscript will be satisfactory.

Comment: 1) abstract: the aim is too technical, can the authors explain the aim at the "research" level?

Authors’ reply: As suggested we have better explained the aim in the abstract, as follows: “The present study aims to investigate the behavioral outcomes and the antecedents’ brain dynamics during the preparation of tasks in which the discrimination is either about the choice (CRT) or the action (Go/No-go), and in a task not requiring discrimination (SRT).”

Comment: 2), introduction: 1st sentence: the use of the term "choice" together with the example of choosing between cake and pizza let the reviewer expect (value-based) decision-making tasks. however, the study has nothing relevant to value or preference-based choices. this part should be revised.

Authors’ reply: We agree with the Reviewer about the value-based expectation of the proposed decision-making example. We have decided to remove the examples to avoid misunderstanding.

Comment: 3), figure 1: the abbreviations of SRT and CRT are better explained in the legend.

Authors’ reply: We have included the explanation of the abbreviations in the legend, as suggested: “Schematic illustration of the three tasks, namely the simple response time task (SRT), the Go/No-go, and the choice response time task (CRT). The green button corresponds to the response made with the index finger, the red button corresponds to the response made with the middle finger, and the red X corresponds to the response withholding.”

Comment: 4), section 2.3 behavioral data analysis: why responses slower than 1000 ms are considered outliers?

Authors’ reply: In response time (RT) research, RT outliers are typically excluded from statistical analysis to improve the signal-to-noise ratio (Berger and Kiefer, 2021). Indeed, the mean RT + the standard deviation of the slowest task (the CRT with middle finger response) was 640 ms. Therefore, it is very probable that the responses so late were conditioned by the participants' inattention, also confusing the interpretation of brain correlates.

Comment: 5), tables 1 and 2: the statistical results reported in the main text are better also presented in the table, which is far easier to read.

Authors’ reply: As suggested by the Reviewer, in the revised version of the manuscript we have reported the main significant statistical effects in tables.

Reviewer 2 Report

Comments and Suggestions for Authors

This is an interesting study examining the electrophysiological correlates of different proactive controls during response competition and inhibition tasks. I agree that it may contribute to the literature. However, I have several concerns regarding the current manuscript:

1. One of my biggest concerns is the inappropriate self-citation by authors. Approximately more than 40% of the citations are self-citations. Anything more than 15% is already a red flag. I don't think this is appropriate at all. Without addressing this problem, it may affect Brain Sciences and MDPI reputation in a long run.

2. More information regarding the inclusion and exclusion criteria of the sample should be reported. There should also be information regarding how participants were recruited.

3. Figure 1 was missing in the manuscript

4. The authors should report the reliability of their cognitive tasks (e.g., odd-even split half reliability). This is to ensure that the correlations reported are robust.

5. It will be useful for the authors to supplement more information on how the counterbalancing was conducted.

6. A sample size of 20 is considered small in this type of study. It will be good for the authors to acknowledge this in their limitation.

Author Response

Comment: This is an interesting study examining the electrophysiological correlates of different proactive controls during response competition and inhibition tasks. I agree that it may contribute to the literature. However, I have several concerns regarding the current manuscript:

Authors’ reply: We thank the Reviewer for the positive consideration of the manuscript. We have addressed all the points raised by the Reviewer, hoping that the revised version of the manuscript will be satisfactory.

Comment: 1. One of my biggest concerns is the inappropriate self-citation by authors. Approximately more than 40% of the citations are self-citations. Anything more than 15% is already a red flag. I don't think this is appropriate at all. Without addressing this problem, it may affect Brain Sciences and MDPI reputation in a long run.

Authors’ reply:  We understand the Reviewer’s concern about the tendency for the work to become self-referential and, for this reason, we removed more than half of the self-citations. Now the percentage is about 24%.

Comment: 2. More information regarding the inclusion and exclusion criteria of the sample should be reported. There should also be information regarding how participants were recruited.

Authors’ reply: We have included a sentence with information of the recruitment of participants and inclusion/exclusion criteria, as follows: “Thus, a total of 20 young (mean age ±SD: 22.7 ±3.1 years) healthy participants (6 females) were involved in the study. All the participants were students at the University of Rome “Foro Italico” and they were given an extra credit for their participation at the study. Inclusion criteria were the following: right-handed [59], corrected-to-normal or normal vision. Exclusion criteria were the following: presence of neuropsychiatric disease, psychological disorders, and neurocognitive drug therapies. All of them provided written informed consent after a full explanation of the procedure by the experimenter, following the Declaration of Helsinki guidelines. The study was approved by the ethical committee of IRCCS Fondazione Santa Lucia (Rome, Italy).”

Comment: 3. Figure 1 was missing in the manuscript

Authors’ reply: Figure 1 was at line 201 (373 in the current version). Reference to Figure 1 was at line 176 (334 in the current version).

Comment: 4. The authors should report the reliability of their cognitive tasks (e.g., odd-even split half reliability). This is to ensure that the correlations reported are robust.

Authors’ reply: As suggested by the Reviewer, we have reported in the current version of the manuscript the odd-even split half reliability test results, showing high and significant correlations within each cognitive task.

Comment: 5. It will be useful for the authors to supplement more information on how the counterbalancing was conducted.

Authors’ reply: The counterbalancing was conducted by means of a balanced Latin square design. This information has been included in the main text, as follows: “the order of tasks was counterbalanced between subjects using a balanced Latin square design to reduce the carryover effect of the within-subject design [60].”

Comment: 6. A sample size of 20 is considered small in this type of study. It will be good for the authors to acknowledge this in their limitation.

Authors’ reply: As suggested by the Reviewer, we included a sentence acknowledging the sample size as a potential limitation of the study, as follows: “Finally, we acknowledge the limited sample size as a potential limitation of the study”.

Round 2

Reviewer 2 Report

Comments and Suggestions for Authors

I think the authors should at least try to reduce another 1 or 2 citations so that it will be at least below 20%. Other than that, the authors have sufficiently addressed my other comments.

Author Response

Comment: I think the authors should at least try to reduce another 1 or 2 citations so that it will be at least below 20%. Other than that, the authors have sufficiently addressed my other comments.

Authors’ reply: We thank the Reviewer for the positive consideration of the revised manuscript. According the suggestion, we have replaced two self-citations with other citations, reacheding the 20%.